# Simulation of Fluid Flow during Egg Pickling under Different Inlet and Outlet Conditions in a Pulsed Pressure Tank with Liquid Circulation

**DOI:** 10.3390/foods11172630

**Published:** 2022-08-30

**Authors:** Jing-Shou Zhang, Magdalena Zielinska, Hui Wang, Yu-Qi Liu, Yu-Fan Xu, Hong-Wei Xiao

**Affiliations:** 1College of Engineering, China Agricultural University, 17 Qinghua Donglu, Beijing 100083, China; 2Department of Systems Engineering, University of Warmia and Mazury in Olsztyn, 10-957 Olsztyn, Poland

**Keywords:** salted egg, pickling tank, computational fluid dynamics, flow velocity distribution

## Abstract

Pulsed pressure pickling is an emerging highly efficient osmotic dehydration technique. However, the immobility of the pickling liquid and the material, the formation of layers, and the uneven pickling efficiency in different sections make it difficult to use industrially. This work aims at improving and optimizing the conditions of fluid flow in the pickling tank with a liquid-cycle system to reduce the unevenness in the production process. Fluid flow around the eggs was numerically investigated by solving three-dimensional Reynolds-averaged Navier–Stokes equations within the flow regime, adopting different angles and positions at the inlet and outlet. The simulation results show that the inlet with a radial deflection of 35° and the outlet with axial direction were characterized by the best flow efficiency. Under these conditions, the average flow velocity and the global uniformity index were 0.153 m/s and 0.407, respectively. Furthermore, the experiments were carried out using an equivalent scale model of the pulsed pressure equipment with liquid circulation. The experimental results showed that, under optimal conditions, the salt content in all four layers of the egg white was about 2.8% after 48 h. This study provides a solution to ensure the constant salinity of different layers of pickled eggs and to improve pickling efficiency, especially in industrial-scale production.

## 1. Introduction

Pickling food in salt or sugar solution is one of the oldest ways to preserve biological materials. Pickling has been often employed to preserve fruits, vegetables, meats, fish, eggs, etc., to extend the shelf life of seasonal foods and reduce postharvest waste [1]. The pickling operation is the process of partially removing water from food by soaking a water-rich solid food in a concentrated hypertonic solution (salt or sugar solutions). It is a counter-current mass transfer process in which water transfers to the surrounding solution and the solute enters the product to reduce the water activity of the food and, thus, inhibit the growth of food spoilage bacteria and prevent food spoilage [2,3]. In addition, pickling can improve the flavor and taste of food, as well as the quality of food products, and minimize the degradation of heat-sensitive bioactive compounds. Generally, pickling is a slow process, dependent on the properties of hypertonic solution, the solution-to-material ratio, and material properties (i.e., shape, size, and permeability of cell membranes) [4,5].

Traditional pickled foods enrich consumers’ sensory perception, providing a wide variety of flavors, colors, and textures. Pickled products, such as pickled cucumbers or salted eggs, are very popular all over the world [6]. However, traditional pickled foods are typically homemade products obtained by spontaneous osmotic dehydration [7]. Traditional natural pickling is time-consuming, labor-intensive, inefficient, and susceptible to environmental factors and harmful microflora [8]. For example, the egg is conventionally covered with a pasty mixture of soil, salt, and water or is immersed in a salt-containing solution for 20–30 days, while garlic pickling takes one to two months to obtain the desired taste [9,10]. The traditional natural pickling process usually lacks hygienic quality control and poses potential safety hazards. For example, the total colony number in naturally pickled chicken reaches 2.71 × 10^4^ CFU/g after 15 days of storage [11]. Nowadays, consumers demand high-quality, nutritious, convenient, additive-free, safe foods with a natural flavor and taste, and an extended shelf-life [12]. To satisfy these consumption trends, the natural technique of osmotic dehydration needs to be replaced by an efficient alternative pickling technology [6,7].

Pulsed pressure osmotic dehydration (PPOD) is a new technology used to improve the mass transfer and quality attributes of different food products [9,13,14,15,16]. PPOD uses high pressure to improve the penetration of the solvent into the product and the low pressure to release the microbubbles from the product tissue into the solution [9]. Due to the synergistic effect of high pressure and atmospheric pressure, PPOD can significantly improve the pickling rate compared to traditional natural osmotic dehydration [17]. PPOD increases the water loss and salt uptake by eggs. The mass transfer was even more than 15 times higher than that noted for the traditional pickling process [10]. In addition, PPOD was able to improve the acetic acid transfer efficiency of garlic and the quality of the final product. Compared to natural osmotic dehydration, the pulsed pressure pickling successfully enhanced the transfer of acetic acid from 0.32 to 0.65%, decreased the degradation of thiosulfinates from 5.18 to 1.72 mg/mL, and minimized the color changes in L* parameter from 81.53 to 38.61, a* parameter from −19.75 to −1.66, and b* parameter from 21.03 to 2.23 of “Laba” garlic [9].

In the industrial practice of pickling eggs, uneven pickling and product quality cannot be guaranteed, especially when spices are added during the pickling process [17,18]. The stratification of the osmotic solution during pickling can lead to a lower concentration of the upper solution than that of the bottom solution, and, thus, lead to an uneven pickling process. In addition, the solution and the material is relatively static during the pickling process, and the solution around the material is easily diluted. This can lead to a reduction in mass transfer rate. Liquid-cycle pickling can solve the problem of unevenness, enhance the pickling rate, and improve the quality of the final product [18,19]. However, fluid circulation during pickling under different inlet and outlet conditions, particularly different positions, may have a great influence on the pickling efficiency. Therefore, determining the position of the inlet and outlet is a key issue in the design of liquid-cycle pickling equipment.

Many studies have been carried out in agricultural ventilation systems to experimentally and/or numerically characterize the influence of aerodynamic and geometric parameters related to the inlet nozzle, dimensions, and the position of the inlet and outlet sections on the airflow course and ventilation efficiency [20,21,22,23,24]. Benni et al. compared the roof vent opening configurations by computational fluid dynamics (CFD) simulations to optimize ventilation in the greenhouse [25]. Cheng et al. studied the natural ventilation rates and airflow patterns in multi-span greenhouses and glass greenhouses, respectively. They found a good balance between airflow and wind speed [20]. CFD has emerged as an important approach to investigate flow, heat, and mass transfer processes in various industrial application scenarios, such as product stacking, can sterilization, fruit and vegetable drying, and others [26]. Due to the high cost of commercial-scale experiments, numerical models are widely used to analyze or improve existing processes and designs [26,27].

Therefore, this study aimed to investigate fluid flow during egg pickling under different inlet and outlet conditions in a pulsed pressure tank with liquid circulation using the computational fluid dynamics method. This work will contribute to a better understanding of flow behavior under different conditions, which is essential for optimizing the design of the liquid circulation in PPOD.

## 2. Materials and Methods

The behavior and flow during pickling of eggs are modelled using a CFD tool FLUENT 2019 (ANSYS Inc., Canonsburg, PA, USA). The subsequent sections discuss the mathematical model, the detailed validation, and the numerical setup used in this study.

### 2.1. Mathematical Model

The Reynolds-averaged Navier–Stokes equations (RANS equations) are time-averaged equations of motion for fluid flow. The idea behind the equations is Reynolds decomposition, whereby an instantaneous quantity is decomposed into its time-averaged and fluctuating quantities. The RANS equations are primarily used to describe turbulent flows. These equations can be used with approximations based on knowledge of the properties of flow turbulence to give approximate time-averaged solutions to the Navier–Stokes equations. The Reynolds-averaged Navier–Stokes equations (RANS) were solved using the commercial ANSYS FLUENT code, the purpose of which was to simulate the flow around the eggs. The theoretical model that consisted of the general equations of conservation of mass and momentum was expressed by Equations (1) and (2):(1)$$\frac{\partial (\rho u_{i})}{\partial x_{i}}=0$$
(2)$$\frac{\partial \left(\rho u_{i}u_{j}\right)}{\partial x_{j}}=-\frac{\partial p}{\partial x_{i}}+\frac{\partial }{\partial x_{j}}\left(\mu \frac{\partial u_{i}}{\partial x_{j}}-\rho \bar{u_{i}^{\prime }u_{j}^{\prime }}\right)$$
where *p*, *μ*, *u_i_*, and $u^{\prime }$ indicate static pressure, the dynamic viscosity of the fluid, average velocity component (*x*, *y*, *z*), and fluctuation velocity, respectively, while *i*, *j* = 1, 2, 3 (*x*, *y*, *z*).

In Equation (2), there are $-\rho \bar{u_{i}^{\prime }u_{j}^{\prime }}$ unknown Reynolds stresses resulting from the distribution of instantaneous velocities into mean and fluctuating components. To construct a closed-form solution, the standard *k–ε* model was based on the closure of the Reynolds stress equation.

Boussinesq’s hypothesis is that the turbulent stresses are related to the mean velocity gradients in almost the same way that the viscous stresses are related to the complete velocity gradients. Based on the Boussinesq hypothesis, the Reynolds stresses can be described as follows [28]:(3)$$-\rho \bar{u_{i}^{\prime }u_{j}^{\prime }}=-\mu_{t}\left(\frac{\partial u_{i}}{\partial x_{j}}+\frac{\partial u_{j}}{\partial x_{i}}\right)+\frac{2}{3}\rho \delta_{ij}k$$

The turbulent (eddy) viscosity $\mu_{t}$ can be obtained from the following equation [29]:(4)$$\mu_{t}=\rho C_{\mu }\frac{k^{2}}{\varepsilon }$$
where the adjustable constant *C_μ_* = 0.09 in the standard *k–ε* model.

Turbulence was predicted with the standard *k–ε* turbulence model [30]. The equations describing the standard *k–ε* model are as follows:

*k* equation
(5)$$\frac{\partial \left(\rho ku_{i}\right)}{\partial x_{i}}=\frac{\partial }{\partial x_{i}}\left[\left(\mu +\frac{\mu_{t}}{\sigma_{k}}\right)\frac{\partial k}{\partial x_{i}}\right]+P_{k}-\rho \varepsilon$$

*ε* equation
(6)$$\frac{\partial \left(\rho \varepsilon u_{i}\right)}{\partial x_{i}}=\frac{\partial }{\partial x_{i}}\left[\left(\mu +\frac{\mu_{t}}{\sigma_{\varepsilon }}\right)\frac{\partial \varepsilon }{\partial x_{i}}\right]+\frac{\varepsilon }{k}\left(C_{1}P_{k}-C_{2}\varepsilon \right)$$
where the empirical constants $\sigma_{k}$ = 1.0, $\sigma_{\varepsilon }$ = 1.33, *C*_1_ = 1.44, and *C*_2_ = 1.92 in the standard *k–ε* model.

The production rate of turbulent kinetic energy *P**_k_* in Equation (7) is given as follows [31]:(7)$$P_{k}=\mu_{t}\left(\frac{\partial u_{i}}{\partial x_{j}}+\frac{\partial u_{j}}{\partial x_{i}}\right)\frac{\partial u_{i}}{\partial x_{j}}$$

In this numerical model, uniform distribution is assumed for velocity components at the inlet, kinetic energy of turbulence *k*_0_, and the energy dissipation rate, *ε*_0_. The numerical values are specified as [32,33]:(8)$$k_{0}=\frac{3}{2}{\left(U_{0}I_{0z}\right)}^{2}$$
where *I*_0*z*_ represents the turbulence intensity of the z-component of velocity at the inlet.
(9)$$\varepsilon_{0}=\frac{C_{\mu }^{0.75}k_{0}^{1.5}}{0.07D_{H}}$$
where *D*_H_ represents the hydraulic diameter of the inlet section.

### 2.2. Description of the Simulation Cases

In this experiment, water was used to prepare a model aqueous salt solution. The flow of fluid in the vessel was forced by a centrifugal pump (Lansheng Pump Technology Inc., Suzhou, Jiangsu, China). The fluid circulated between the inlet and outlet. The inlet flow rate and outlet pressure were set at 1.13 m/s and 0 Pa, respectively. A few different cases were distinguished mainly by the different angles between the inlet and outlet (Table 1). In all cases, the inner diameter of the inlet and outlet was 25 mm, and the distance between the inlet and outlet was 365 mm (except for the axial outlet). In these cases, four egg layers have been distinguished, and the distance between the middle sections of the egg layers was 90 mm (Figure 1).

The positions and angles of the inlet and outlet are shown in Figure 2. Firstly, the inlet was placed in a fixed position and the outlet was located at a constant angle to the inlet. The inlet and outlet were oriented in a radial direction (Figure 2a). Then, the inlet and outlet were inclined by 35 and 55°, respectively, in opposite directions from the radial direction (Figure 2b). The next system used an axial outlet and the radial inlet in the lower middle part of the pickling tank (Figure 2c). Finally, the system used an inlet with a radial deflection of 35° and the outlet in the center of the lower part of the pickling tank (Figure 2d).

### 2.3. Boundary Conditions and Computational Meshes

The commercial CFD software package (ANSYS Inc., Canonsburg, PA, USA), MESHING and FLUENT 19.0, was used to create three-dimensional (3D) geometry, generate grids, and solve the three-dimensional turbulence model. The physical picture of the pulsed pressure tank is shown in Figure 3a. A full-scale computational domain with a length of 300 mm and a height of 565 mm is shown in Figure 3b. The co-ordinate system for the numerical model was a right-handed 3D Cartesian co-ordinate system. The co-ordinate origin was set to the center of the free surface mesh plane. The z co-ordinate in the co-ordinate system was negative in the flow direction, the x co-ordinate was perpendicular to the horizontal direction of flow, and the y co-ordinate was negative towards the acceleration due to gravity. The upper boundary of the convex slot was described by the velocity–inlet boundary condition, while the lower boundary was described by the outflow boundary condition. Velocity distribution at the inlet was assumed uniform [34]. The slot and the free surfaces were modeled using the wall boundary condition equal to zero shear force. All the walls of the fluid are assumed to be adiabatic. In the proceeding analysis, all fluids are assumed to be isotropic and Newtonian, and the flow is incompressible. For simulation purposes, unstructured meshes with a predominance of hexahedral computational cells were produced (Figure 3c), and the maximum size of the cells in the domain was set to 5 mm. In the regions close to the spheroid surfaces, the techniques of wall function [35] and boundary-layer mesh were adopted to solve the effect of the solid surfaces [36,37]. In the boundary area, five boundary layers with an expansion ratio of 1.2 were generated (Figure 3d).

### 2.4. Numerical Algorithm Method

The three-dimension numerical model was divided into cells, while the governing equations were discretized using the finite volume method and solved with the CFD Fluent 2019 software, ANSYS Inc. The governing equations were solved with a 3D pressure-based solver. The solution loop was carried out iteratively to obtain a converged numerical solution. SIMPLEC solution algorithms for pressure–velocity coupling were used to solve steady problems in CFD. The discretization scheme for pressure, momentum, turbulent kinetic energy, and turbulent dissipation rate were carried out using a second-order upwind scheme. The upwind scheme referred to a numerical discretization method for solving hyperbolic partial differential equations, in which upstream variables were used to calculate the derivatives in a flow field. It means that derivatives were estimated using a set of data points biased to be more “upwind” of the query point, concerning the direction of the flow. Convergence was considered to have been reached when the residuals were less than 10^−4^ for the flow variables (continuity, *x*-, *y*-, and *z*-velocities, *k*, and *ε*) and 1000 iterations were required to reach steady state.

### 2.5. Assessment Criteria

Definition of the sectional uniformity index

The uniformity index for sectional velocity *M_f_* was defined as [38]:(10)$$\;M_{f}=\sqrt{\frac{1}{n}\sum_{i=1}^{n}{\left(\frac{u_{i}-\bar{u}}{\bar{u}}\right)}^{2}}$$
where *n* indicates the number of nodes on the plane, *u_i_* indicates the velocity magnitude at node *i*, and indicates the average velocity on the selected face.

Characterization of the global uniformity index

Wu [39] proposed the uniformity index for global velocity (UI value):(11)$$\mathrm{UI}=\frac{\sum_{i=1}^{n}\left(\left|v_{i}-\bar{\nu }\right|\cdot V_{i}\right)}{\sum_{i=1}^{n}\left(v_{i}\cdot V_{i}\right)}$$
where *n* indicates the number of mesh cells, *v_i_* indicates the velocity magnitude at cell *i*, $\bar{v}$ indicates the average velocity throughout the whole tank, and *V_i_* indicates the volume at cell *i*.

### 2.6. Grid Independence Verification

To make sure that the results are grid-independent, a grid independence study was conducted. The significance of these figures resides in their ability to make it evident that all the examined grids can capture the counter-rotating vortices. To obtain the optimal mesh type and mesh size, the boundary layer around the egg and the wall was encrypted by selecting the better case i. To quantify the dependence of the numerical results on the cell density, numerical simulations were performed for four grids with different cell densities (Table 2). The three densest meshes calculated the sectional uniformity index (Section 1) within a 0.7% range of their average, indicating that normal mesh provided an adequate level of accuracy. In particular, the final mesh (Normal) is reported in Table 2.

## 3. Results and Discussion

### 3.1. Analysis of Velocity Contours of Liquid

The fluid conditions near each egg layer and between the different egg layers were investigated. Figure 4 shows that the circulation of the liquid under different inlet and outlet conditions differed significantly, especially for a radial inlet (cases a–c, h) and an inlet with radial deflection angle (cases d–g, i). The differences concerned mainly the velocity values and velocity distribution in individual cross-sections of egg layer. For example, the mean value of the velocity of the cross-sections of the egg layer in the case of f was approximately twice as high as in the case of c (Figure 5). The inhomogeneity of the velocity in the cross-sections of the fourth egg layer could be seen mainly near the center of the tank. In the case of c, the velocity values for the egg cross-sections were significantly different, and the velocity value of the fourth layer was close to 0. In the case of i, the velocity of the liquid in the middle cross-section of the egg layer has improved significantly, reaching the maximum values of the velocity in the cross-section of the egg layer compared to those achieved in cases of c and f. The abovementioned results are identical to a previous study, e.g., as the liquid flows downward in the tank, the maximum liquid flow velocity decreases and the velocity distribution in the cross-section tends to become a developed flow [40].

### 3.2. Analysis of the Sectional Uniformity Index

The average velocity in the middle of each egg layer was predicted by integrating the velocity magnitude in the selected cross-section domain (Table 3). The calculation of *M_f_* was carried out using the Fluent 19.0 custom field function. The lower the value of the *M_f_*, the more homogeneous the fluid flow distribution is [38]. The results show that the flow velocity inhomogeneity decreases from Section 1 to Section 4. The exception was case h. This could be due to the effect of the liquid projecting from the inlet onto the wall and obstruction to the flow of liquid through the egg layer, reducing the velocity of the liquid in the longitudinal direction. The results also show that the best results were achieved for case i in terms of flow velocity uniformity.

### 3.3. Analysis of the Resultant Velocity Value

The distribution of the resultant velocity of the liquid at different heights of the pickling tank in the cases of a, b, c, d, e, f, g, h, and i is shown in Figure 5. The radial deflection of the inlet in the cases of d, e, f, g, and i exceeded the velocity magnitude and axial velocity fluctuations of the radial inlet of cases a, b, c, and h (Figure 5). For the radial inlet of the liquid stream into the tank, the radial component of the fluid velocity quickly dropped to 0 due to the significant effect on the tank wall. Moreover, the axial component of the fluid velocity also declined rapidly. It can also be seen from the change in the total velocity in the axial direction. More specifically, for the cases of a and d, the resultant velocities were about 0.05 and 0.1 m/s (y = 0.35 m or less). The resultant velocity fluctuation in the case of d was smaller than that in the case of a. In the case of a, the velocity gradually decreased from 0.075 to 0.025 m/s. Previous studies have shown that rotational flow not only destroys the fluid boundary layer, increasing the fluid velocity near the wall, but also changes the fluid flow structure. This causes the fluid to have a large tangential velocity component, causing the movement of the fluid mass in the channel to be subjected to centrifugal force. As a result, turbulent mixing between boundary layer fluid and mainstream fluid was enhanced. The results are consistent with the literature data [41,42].

### 3.4. Analysis of the Centerline Speed Value

The predicted flow rate along the axis reflects the flow rate in the center of the pickling tank, and the flow rate along the center line of a–i is shown in Figure 6. The axial fluctuation of the flow rate in the cases of d, e, f, and g was found to be smaller than that in other cases. It can be due to the bidirectional deflection of the inlet and outlet fluid flow along the tank wall. Then, it may result in the formation of a certain stagnation of fluid in the central part of pickling tank, as shown in Figure 4. The tendency variation in axial velocity in case i could be due to the interaction between the radially deflected inlet, which allowed the rotational flow of fluid through the tank, and the axial outlet at the bottom, which prevented the fluid near the center from stagnating. In addition, the fluid outflow at the bottom also prevented fluid stagnation near the center of the tank, especially near the center of the bottom. In case i, reducing the flow velocity along the axis could result in a stagnation in the center of the vortex. The results are consistent with the literature data [41].

### 3.5. Analysis of Three-Dimensional Isosurfaces and Stream Traces Colored by Velocity Magnitude

Figure 7 shows three-dimensional isosurfaces and stream traces colored by velocity magnitude for four common cases (c, f, h, and i). An isosurface was a surface that represented points of a constant value within a volume of space. In other words, it was a level set of a continuous function whose domain was 3D space. It allowed visualization of the constant value of a contour variable as a surface. When a velocity value was specified in this test, the isosurface showed where the specified variable had this velocity value. Three different velocity values were set in the present study to generate three equivalent surfaces. The resulting three-dimensional isosurfaces are shown in Figure 7. The velocity flow stream shows the formation of a large number of irregular swirls of fluid in the tank, causing the fluid to stagnate in the swirl zone and possible uneven fluid flow (Figure 7a). It is supported by the results shown in Figure 7c. However, Figure 7c shows a large area of stagnation from the fourth egg layer to the outlet, with the flow lines close along the wall towards the outlet, explaining the greatest inhomogeneity where Mf4 was 71.31% (Table 3). As the partial return of the flow velocity back towards the inlet was evident, the entire flow field observed for the case of h was found to be the most inhomogeneous (UI = 0.686). The central velocity contour plot (Figure 7b) shows the formation of stagnation areas near the center of the pickling tank over the entire flow field, consistent with the results shown in Figure 4 and Figure 5. According to the central velocity contour plot (Figure 7d), stagnation of the fluid in the central region due to vortex formation by fluid rotation was significantly improved compared to that shown in Figure 7b. At the same time, the plot of the isosurfaces shows that the mean flow rate of the flow field shown in Figure 7d was larger than that shown in Figure 7b. It further shows that the axial discharge from the bottom of the tank improved the stagnation of the central fluid and increased the magnitude level of the velocity of the flow field in the case of a radially inclined inlet flow, as assumed in Figure 5. The results show that rotational flow in a cylindrical tank contributed to the reduction of a dead zone. The results are consistent with the literature data [43].

### 3.6. Analysis of the Global Uniformity Index

The average velocity in the pickling tank was predicted by integrating the magnitude of the velocity across the computational domain (Table 4). The lower the value of the UI, the smaller the difference in flow rate is. The flow velocity remained the same in any position of the flow field with UI = 0 [39]. The average velocity in the case of i reached the maximum value, i.e., 0.153 m/s, for the same inlet flow rate. In other words, a minimum inlet flow rate (lowest power) was required for the case of i at the same average flow field velocity. The results show that the fluid flow distribution uniformity in the case of i was the best (UI = 0.407) compared to others.

### 3.7. Analysis of the Average Static Pressure

The impact of the flowing fluid on the egg surface under different working conditions (cases a–i) was assessed in the present study. The mean static pressure of the fluid on the first layer (Section 1) of the egg surface is shown in Figure 8. The turbulent wall jet is known to be a difficult flow to predict. The resulting flow was complicated since it is the combination of free turbulent shear flows, near-wall effect, and recirculation areas (including high streamline curvature and local separation) [44]. It should be noted that the fluid flow inside the container was complex (the hydrostatic pressure exerted on the surface of the eggs may not be uniform). In addition, it is reasonable to focus on all eggs in the region of interest if there is a large number of eggs in the pickling container. Therefore, the mean static pressure was used to reflect the trend of the impact on the surface of the egg layer. The average static pressure of case i was found to be the highest one (Figure 8). The inlet and outlet settings of scheme i were found to be more preferable than that of other schemes.

### 3.8. Validation of Mode

The PPOD equipment was designed and installed at the College of Engineering of China Agriculture University, Beijing, China. It was previously described in detail in the literature [9,10]. To verify the credibility of the simulation, tests were carried out using pickling equipment with pressure pulsation and fluid circulation. For the tests, two selected conditions were chosen, namely h and i. For the control experiment, namely j, the eggs were impregnated in pressure-pulsed pickling tanks with no liquid circulation. The following experimental conditions, i.e., pressure, the temperature of pickling solution, frequency of pressure pulsation, and operating time of centrifugal pump, were set at 120 kPa, 25 °C, 7.5 min/15 min, and 4 min/18.5 min, respectively. The eggs were immobilized on a rack immersed in a saturated salt solution (25% NaCl (*w*/*w*)) at 25 ± 2 °C for up to 48 h. In such an arrangement, four different egg layers have been distinguished. The test results were assessed by measuring the salt and water content of the egg white (EW) and egg yolk (EY). All experiments were repeated in triplicate and mean values were reported.

The salt content of EW/EY of eggs pickled for 48 h is shown in Figure 9. It can be seen that the pickling rate was improved by pulsed pressure pickling with liquid circulation (Figure 9, cases h and i vs. case j). The EW salt content for the four layers (cases i and j) differed slightly after pickling for 48 h, remaining at around 2.8 and 2.1%, respectively. The salt content in the EW of four layers of eggs (case h) varied greatly. The salt content in the EW gradually decreased and the difference between the upper and the lower layer was 0.35% after 48 h of pickling. The EY salt content for layers (cases h, i, and j) differed slightly after pickling for 48 h, remaining at 0.4–0.65%. The results show that different inlet and outlet positions and angles greatly influenced the pickling effect. The water content of EW/EY of eggs pickled for 48 h is shown in Figure 10. It can be seen that the dehydration rate was improved by pulsed pressure pickling with liquid circulation (Figure 10, cases h and i vs. case j). The osmotic dehydration of EY was more obvious than EW. The EY water content for the four layers (cases i and j) differed slightly after pickling for 48 h, remaining at around 35.0 and 43.5%, respectively. The experimental and simulation results show that the unevenness in the pickling rate was closely related to the flow rate of the pickling solution near the eggs. The flow of fluid near the egg surface promoted the migration of water from the inside of the eggs and the breakdown of the boundary layer that was formed near the eggshell. Most likely, PPOD resulted in increased leaching of substances from the egg, forming a concentration boundary layer that prevented salt osmosis [45]. Further, the pickling liquid circulation can break the boundary layer and improve the pickling efficiency. However, a suitable inlet and outlet are needed to ensure the flow of pickling liquid near the egg.

## 4. Conclusions

The study focuses on the evaluation of the flow performance of full-size pickling tanks with nine different inlet and outlet fluid circulation designs using CFD. Qualitative and quantitative data for each scenario provided valuable information on the fluid flow in the pickling tank containing four different egg layers. The results show that the 35° radial deflection pattern of the inlet and lower axial outlet was superior to the other simulation cases in terms of the uniformity of cross-sectional velocity, as well as average and overall velocity. The simulation results for this case shows that the cross-sectional uniformity indices were 0.574, 0.467, 0.394, and 0.319 for Section 1, 2, 3, and 4, respectively, while the global uniformity index and average velocity were 0.407 and 0.153 m/s, respectively. The optimal simulation scheme ensured great pickling results, i.e., the salt content of all four layers of EW at around 2.8%. Experimental results proved that fluid circulation improved pickling efficiency. However, the result can be uneven if the inlet and outlet channels are not properly positioned. Satisfactory results confirmed the rationality of using CFD simulations for this purpose. The results also provide theoretical support for the design of equipment with pressure pulsation and fluid circulation used in the industrial production of uniformly pickled salted eggs.

## Figures and Tables

**Figure 1 foods-11-02630-f001:**
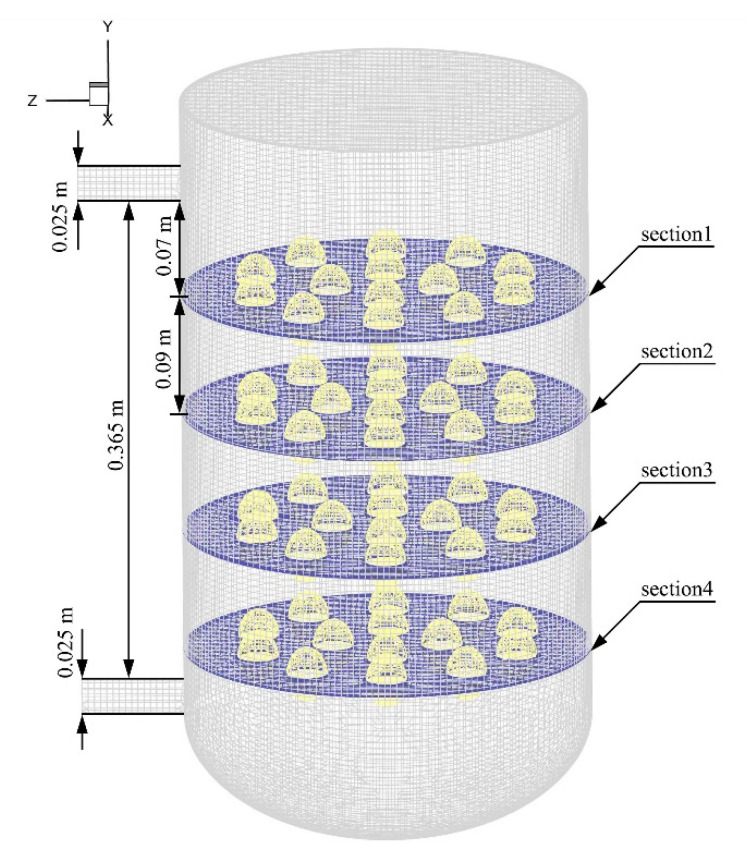
Location of the selected four layers.

**Figure 2 foods-11-02630-f002:**
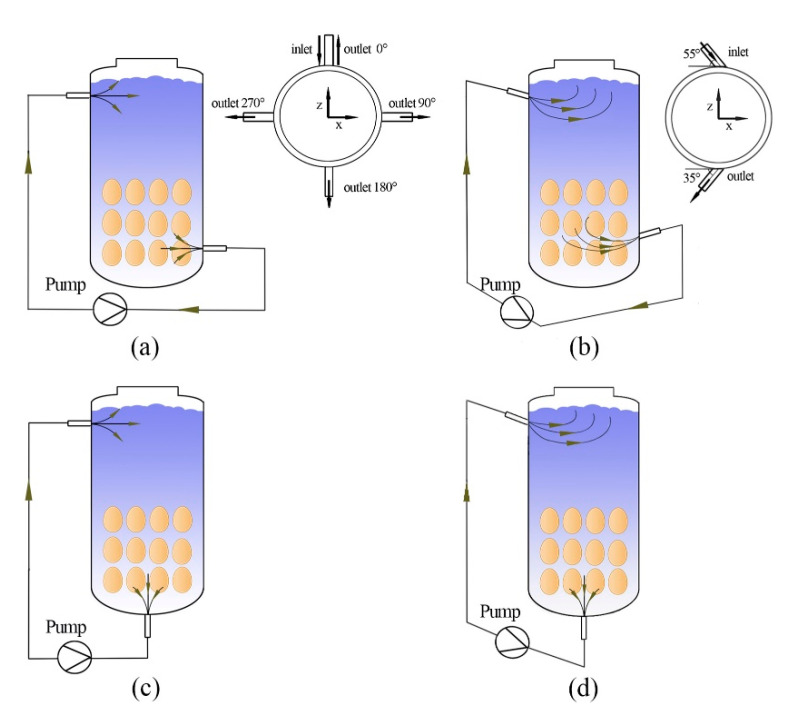
Schematic diagram of the liquid-cycle inlet and outlet settings: (**a**) radial inlet and outlet, (**b**) radial inclination of the inlet and outlet at a certain angle, (**c**) radial inlet and axial outlet, (**d**) radial deflection of the inlet and axial outlet.

**Figure 3 foods-11-02630-f003:**
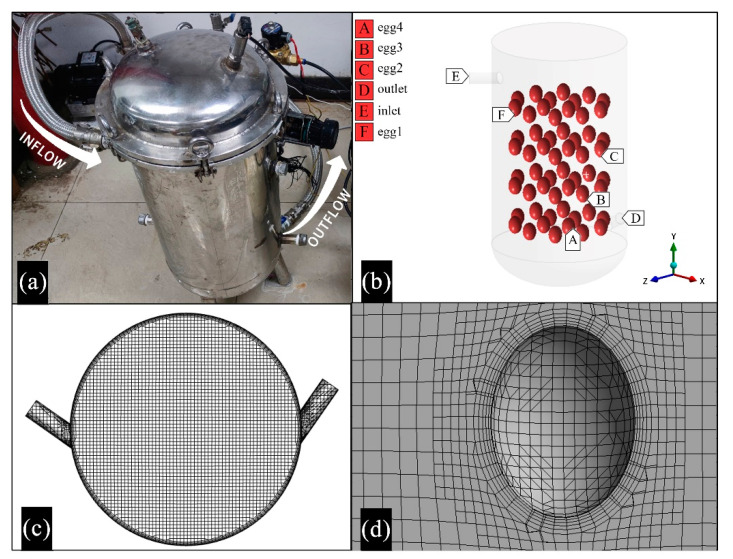
Models and computational meshes: (**a**) material object, (**b**) 3D model, (**c**,**d**) boundary conditions and computational meshes.

**Figure 4 foods-11-02630-f004:**
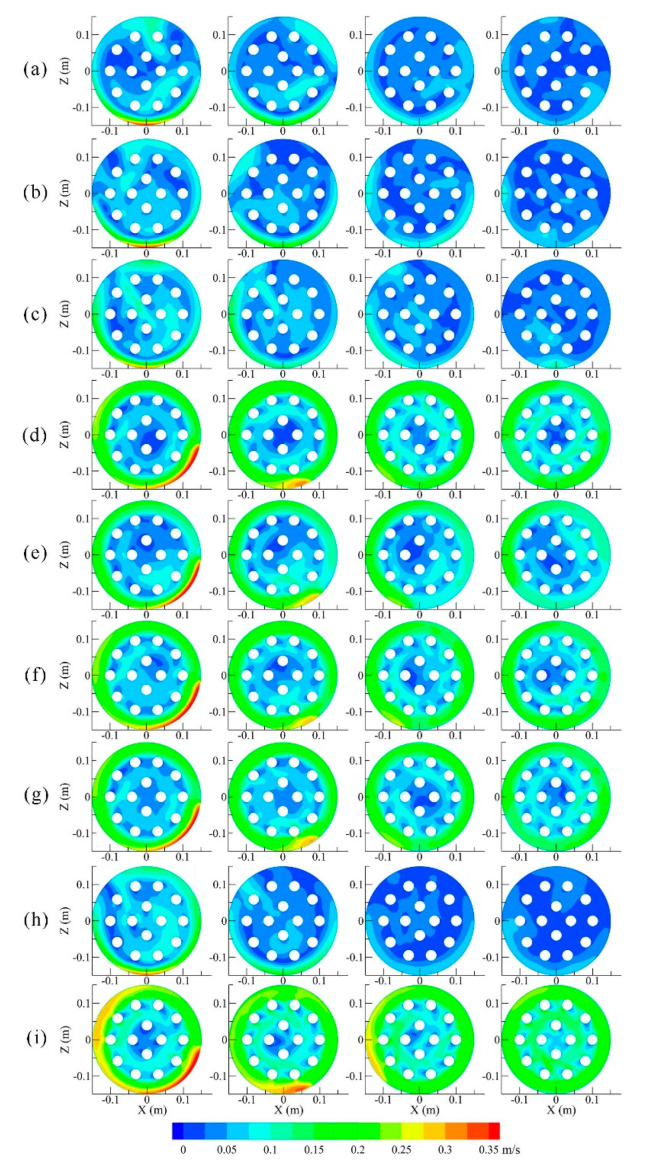
Contours of liquid velocity at sections 1, 2, 3, and 4 under different conditions (**a**–**i**).

**Figure 5 foods-11-02630-f005:**
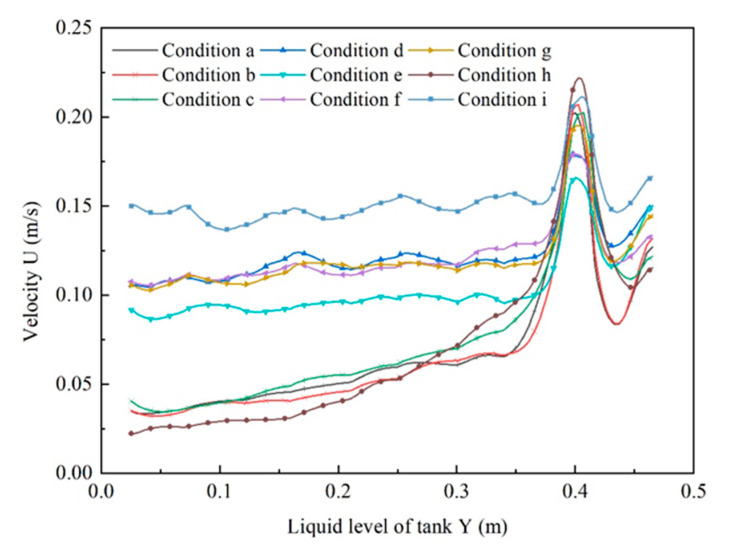
Distribution of liquid resultant velocity value at different heights of pickling container under different conditions (a–i).

**Figure 6 foods-11-02630-f006:**
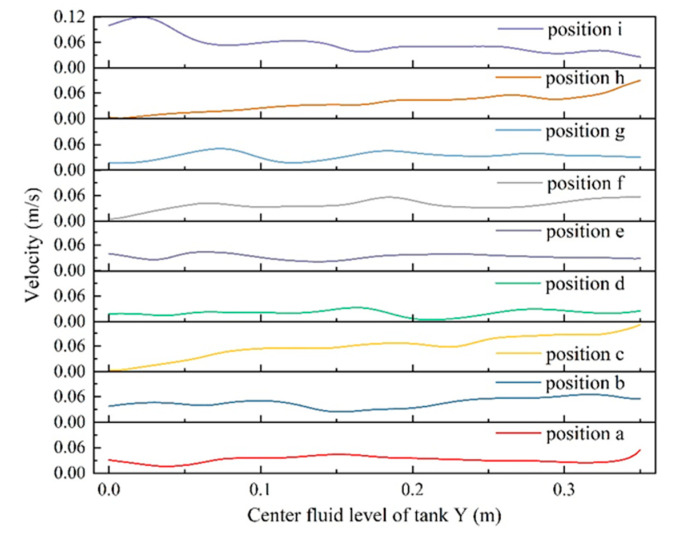
Flow velocity along the a–i line.

**Figure 7 foods-11-02630-f007:**
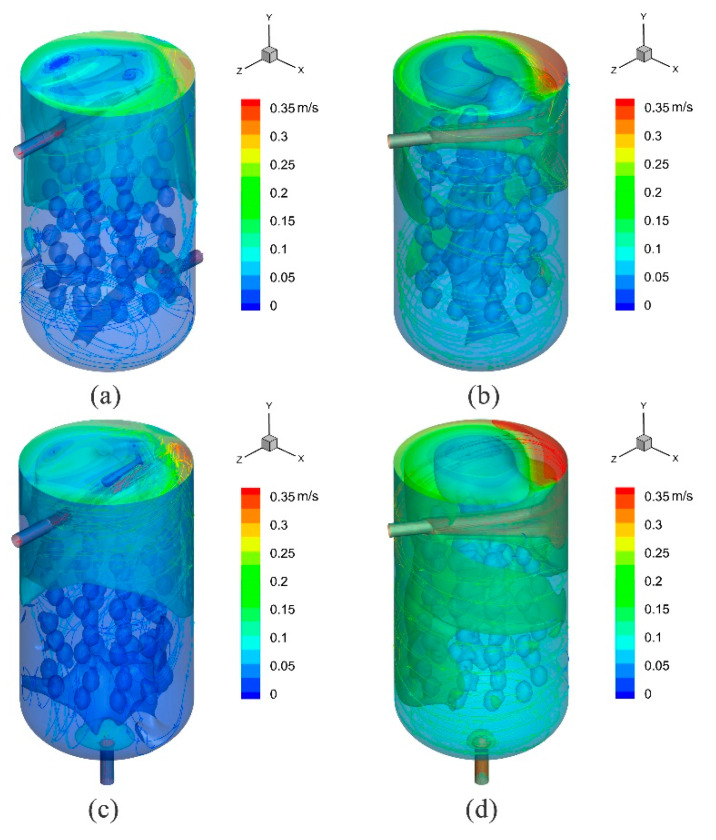
Three-dimensional iso-faces and stream trace colored according to velocity magnitude: (**a**) case c, (**b**) case f, (**c**,**d**) case h and i.

**Figure 8 foods-11-02630-f008:**
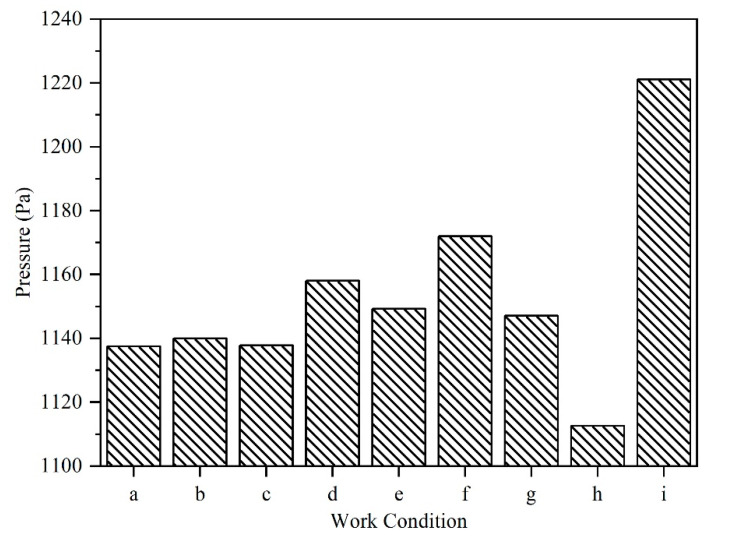
Average pressure on the first layer of eggs under different operating conditions (a–i).

**Figure 9 foods-11-02630-f009:**
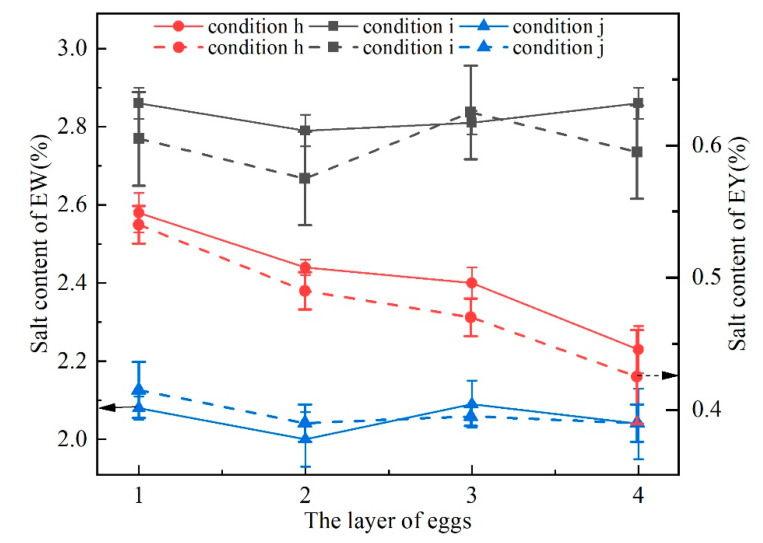
EW/EY salt content in different layers under operating conditions h, i, and j.

**Figure 10 foods-11-02630-f010:**
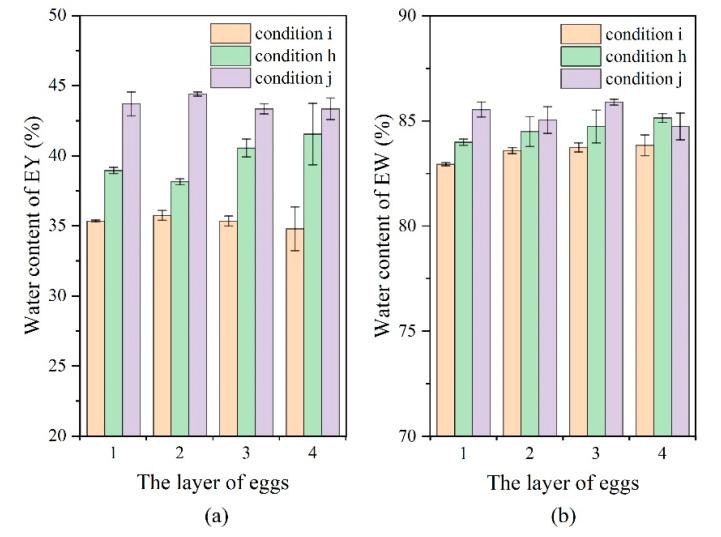
(**a**) EY water content in different layers under operating conditions h, i, and j; (**b**) EW water content in different layers under operating conditions h, i, and j.

**Table 1 foods-11-02630-t001:** Nomenclature of different cases.

Direction	The Angle between Inlet and Outlet				Radial	Deflection
	0°	90°	180°	270°		
Radial	a	b	c	/ *		
Deflection	d	e	f	g		
Axial					h	i

* Radial inlet and outlet with an angle of 270° was calculated almost identically to 90° and was rejected.

**Table 2 foods-11-02630-t002:** Cross-sectional uniformity indices (*M**_f_*_1_) for Section 1 obtained from numerical calculations for different cell densities.

Computational Mesh	Cell Number	*M_f_* _1_
Coarse	714849	53.48%
Normal	989957	57.41%
Dense-1	1170340	57.83%
Dense-2	1292035	58.11%

**Table 3 foods-11-02630-t003:** The *M_f_* values of the egg cross-sections 1, 2, 3, and 4.

Case	*M_f_* _1_	*M_f_* _2_	*M_f_* _3_	*M_f_* _4_
a	0.731	0.568	0.435	0.457
b	0.671	0.639	0.531	0.493
c	0.579	0.493	0.463	0.432
d	0.655	0.566	0.447	0.401
e	0.666	0.551	0.542	0.484
f	0.675	0.523	0.460	0.440
g	0.629	0.529	0.502	0.410
h	0.586	0.525	0.668	0.713
i	0.574	0.467	0.394	0.319

**Table 4 foods-11-02630-t004:** Average velocity and uniformity index for velocity.

Case	$\bar{v}$ (m/s)	UI
a	0.065	0.610
b	0.062	0.622
c	0.072	0.575
d	0.121	0.489
e	0.103	0.497
f	0.120	0.489
g	0.119	0.481
h	0.066	0.682
i	0.153	0.407

## Data Availability

No new data were created or analyzed in this study. Data sharing is not applicable to this article.

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
