# Peer review of "Simulation of Fluid Flow during Egg Pickling under Different Inlet and Outlet Conditions in a Pulsed Pressure Tank with Liquid Circulation"

_foods, 2022, doi:10.3390/foods11172630_

Round 1

Reviewer 1 Report

Review of the manuscript entitled: Simulation of Fluid Flow During Egg Pickling Under Different Inlet and Outlet Conditions in a Pulsed Pressure Tank with Liquid Circulation

The reviewed manuscript is quite interesting. Simulations of the model of the improvement conditions of fluid flow in the pickling tank with a liquid-cycle system are an interesting and relatively difficult approach for correct modeling. Generally, the manuscript is written on the average/low level but contains lots of lacks and elements that need to extend. Below are the main elements:

1. Simulation tool should be presented. Some descriptions and the most important justification of choice should be presented. 

2. Initial conditions and simplifications of the simulation model should be presented and (the most important) discussed.

3. The boundary conditions of the mathematical model (in the mathematical meaning) should be presented (for example check and follow: https://doi.org/10.1016/j.jfoodeng.2019.109846)

4. Details of the mesh quality testing (grid quality parameters) should be provided and discussed. Some discussion is necessary. Density grid testing is inadequate, especially when the authors used the Fluent automatic procedure dedicated for engineering use only. 

5. If the reviewed paper really presents an optimization (in the scientific full meaning of this term) the authors must present typical optimization elements like i.e., goal function, optimization criteria, and their scales. Optimization process methodology is necessary also. “Optimization" it's no WORD for conditions improvement describing - there is a mathematical procedure.

6. The used tool (ANSYS Fluent) has lots of default settings also. The presented case definitely seems a swirled flow (i.e. see types with tangential inflow - check other papers from the above where the tangential inlet case was presented and discussed). What about adopting settings for the considered type of flow?

7. The authors consider the k − epsilon turbulence model. What are the advantages of adopting this approach over others in considering the case? How will this affect the results? The authors should provide more details on this also on the basis of literature research and justify their own choice in the view of the simulated case. Also, coefficients for the chosen model should be presented and discussed in the view of the simulated case, especially when chemical reactions were included. Consider turbulence model is a semi-empirical model, so the coefficients must include experimental conditions of the simulated case. 

8. Details about the convergence of the computing model testing should be provided and discussed.

9. What about comparing CFD and experimental results? Some discussion is necessary, especially on the main subject of simulation. Some extended literature research is necessary. However, some comparisons, especially on the basis of the own experimental results are necessary. It's indispensable.

10. In the end the most important thing. The presented case is definitely a transient case. The process strongly depends on time. Why the authors didn't present and discuss simulation results in the view of the time parameter? The simulation must be done like a transient also.

Also:

11. Change connotation (units) in figs to scientific.

In summary:  I think that the manuscript has some potential and shall be interesting in the future for potential readers, but without very deep rebuilding and the most important experimental comparing results is completely valueless.

Author Response

Reviewer #1:

The reviewed manuscript is quite interesting. Simulations of the model of the improvement conditions of fluid flow in the pickling tank with a liquid-cycle system are an interesting and relatively difficult approach for correct modeling. Generally, the manuscript is written on the average/low level but contains lots of lacks and elements that need to extend. Below are the main elements:

  1. Simulation tool should be presented. Some descriptions and the most important justification of choice should be presented.

Response: Thank you for the good suggestion. We have added the numerical simulation details. The behavior and flow during eggs are modelled using an CFD tool FLUENT 2019. The subsequent sections discuss the mathematical model, the detailed validation, and the numerical setup used in this study.

  1. Initial conditions and simplifications of the simulation model should be presented and (the most important) discussed.

Response: Thanks for your useful feedback. The actual working condition diagram as shown in Figure 3-1 is compared with the simulated geometric model; the size of the simulated geometric model is consistent with the actual working condition. The simplification of the simulation model is also supplemented: the velocity distribution at the inlet is assumed to be uniform; all the walls of the fluid are assumed to be adiabatic; in the proceeding analysis, all fluids are assumed to be isotropic and Newtonian and, the flow is incompressible; in the regions close to the spheroid surfaces, the techniques of wall function and boundary-layer mesh were adopted to solve the effect of the solid surfaces.

  1. The boundary conditions of the mathematical model (in the mathematical meaning) should be presented (for example check and follow: https://doi.org/10.1016/j.jfoodeng.2019.109846)

Response: Thanks very much for the professional comment. Velocity distribution at the inlet was assumed uniform. The inlet flow rate and outlet pressure were set at 1.13 m/s and 0 Pa. The addition of the mathematical model is as follows. In this numerical model, uniform distribution is assumed for velocity components at the inlet, kinetic energy of turbulence k0 and the energy dissipation rate ε0 The numerical values are specified as Eqs.(8) and (9).

  1. Details of the mesh quality testing (grid quality parameters) should be provided and discussed. Some discussion is necessary. Density grid testing is inadequate, especially when the authors used the Fluent automatic procedure dedicated for engineering use only.

Response: Thanks for your suggestion. A description of grid settings has been added. For example: The grid growth rate is 1.2, and the methods of capturing curvature and capturing neighboring grids are used. In addition, a description of grid density (grid independence test) has been added. To make sure that the results are grid independent, a grid independence study was conducted. The significance of these figures resides in their ability to make it evident that all the examined grids can capture the counter-rotating vortices. In particular, the final mesh (Normal) is reported in Table 2. The figure below shows that the average element quality of these groups of grids is above 0.7.

  1. If the reviewed paper really presents an optimization (in the scientific full meaning of this term) the authors must present typical optimization elements like i.e., goal function, optimization criteria, and their scales. Optimization process methodology is necessary also. “Optimization" it's no WORD for conditions improvement describing - there is a mathematical procedure.

Response: Thanks very much. It is true that this study may not have optimized the conditions. This study mainly explores the regularity study under different working conditions. The innovation lies in the use of CFD simulation method, which can explain the reasons of some actual phenomena from the perspective of fluid mechanics, and solve the problems in certain conditions. Under a specific working condition, some parameter settings are more reasonable. If it is described from the perspective of optimization, it may be lack of logic. Because for optimization, it may be the impact of the composite effect of working condition parameters and geometric model parameters on a single or multiple indicators. There are tools for fluid optimization in ansys software, which are further divided into topology optimization and parameter optimization accompanying the solution.

  1. The used tool (ANSYS Fluent) has lots of default settings also. The presented case definitely seems a swirled flow (i.e. see types with tangential inflow - check other papers from the above where the tangential inlet case was presented and discussed). What about adopting settings for the considered type of flow?

Response: Thanks to the reviewer for professional comment. Many important engineering flows involve swirl or rotation such as stirring, mixing, burning, impurity removal, etc. There are many related researches and literatures. We have tried from the characteristics of swirl, such as the vorticity analysis under the Q criterion, and compared the simulation results with the relevant literature. But we have poor results in generating 3D vorticity maps. Most importantly, we did not link the analysis of swirls well to our research purposes, which may lead to our next work.

  1. The authors consider the k − epsilon turbulence model. What are the advantages of adopting this approach over others in considering the case? How will this affect the results? The authors should provide more details on this also on the basis of literature research and justify their own choice in the view of the simulated case. Also, coefficients for the chosen model should be presented and discussed in the view of the simulated case, especially when chemical reactions were included. Consider turbulence model is a semi-empirical model, so the coefficients must include experimental conditions of the simulated case.

Response: Thanks for your useful feedback. The two-equation k-epsilon turbulence model was selected which was demonstrated by several authors (Ni et al., 2016; Yang et al., 2014; Gebremedhin et al., 2005; HOU et al., 2008) to be robust and stable for single-phase airflow in tower fluid environments. The relevant references are as follows:

Ni, P., Jonsson, L. T., Ersson, M., & Jönsson, P. G. (2016). A new tundish design to produce a swirling flow in the Sen during continuous casting of steel. Steel Research International, 87(10), 1356–1365

Yang, Y., Jönsson, P. G., Ersson, M., Su, Z., He, J., & Nakajima, K. (2015). The influence of swirl flow on the flow field, temperature field and inclusion behavior when using a half type electromagnetic swirl flow generator in a submerged entry and Mold. Steel Research International, 86(11), 1312–1327.

Gebremedhin, K., & Wu, B. (2005). Simulation of flow field of a ventilated and occupied animal space with different inlet and outlet conditions. Journal of Thermal Biology, 30(5), 343-353.

Hou, Q., Yue, Q., Wang, H., Zou, Z., & Yu, A. (2008). Modelling of inclusion motion and flow patterns in swirling flow tundishes with symmetrical and asymmetrical structures. ISIJ International, 48(6), 787–792.

  1. Details about the convergence of the computing model testing should be provided and discussed.

Response: Thanks. Convergence was considered to have been reached when the residuals were less than 10-4 for the flow variables (continuity, x-, y-, and z-velocities, k, and ε) and 1000 iterations were required to reach steady state.

  1. What about comparing CFD and experimental results? Some discussion is necessary, especially on the main subject of simulation. Some extended literature research is necessary. However, some comparisons, especially on the basis of the own experimental results are necessary. It's indispensable.

Response: Thanks very much for your good question. As comparing CFD and experimental results: According to qualitative and quantitative (speed uniformity index) analysis and comparison of 9 working conditions, working condition h and working condition i are selected, the test equipment is improved and the actual egg pickling test is carried out. The position and direction of the inlet and outlet on the pickling tank were evaluated according to the effect of 4-layer egg curing (salt content of egg white and yolk), and an experimental comparison was also made indirectly to the typical working conditions simulated by CFD. More information is available in section 3.8.

  1. In the end the most important thing. The presented case is definitely a transient case. The process strongly depends on time. Why the authors didn't present and discuss simulation results in the view of the time parameter? The simulation must be done like a transient also.

Response: Thanks for your feedback. In fact, this paper does not involve the chemical reaction process, but only studies the internal flow field distribution, so the effects of steady-state and transient calculations to achieve a steady state are consistent. Also, in Section 3.8, the eggs were immobilized on a rack immersed in a saturated salt solution (25% NaCl (w/w)) at 25 ± 2 ℃ for up to 48 h. The flow field inside the tank reaches a steady state in a very short time.

Also:

  1. Change connotation (units) in figs to scientific.

In summary: I think that the manuscript has some potential and shall be interesting in the future for potential readers, but without very deep rebuilding and the most important experimental comparing results is completely valueless.

Response: We are very sorry for some mistake that we made in this manuscript. We have checked and revised it.

Reviewer 2 Report

The manuscript 'Simulation of Fluid Flow During Egg Pickling Under Different Inlet and Outlet Conditions in a Pulsed Pressure Tank with Liquid Circulation' meets scientific merit. The English is satisfactory and readers will understand its content. The authors provided information on four layers of egg pickling using Computational Fluid Dynamics by describing the algorithms and simulations involved. The research design and methods have been described appropriately. However, to maintain the quality and impact of the Journal, the following comments are necessary for a revision:

1. Introduction: In Lines 83-85, provide the findings of those studies. What were the reported results? It is vital to mention them in the Introduction.

2. In Section 2.3; Figure 3(2) is not mentioned in the text. Please, highlight it in the text. 

3. In Sections 3.4; Figure 6 should be placed at the end of the text in Line 279.

4. In Sections 3.5; Figure 7 should be placed at the end of the text in Line 308. Ensure consistency of text format/style for the mention of Figs. 4 and 5 or Figure 4 and Figure. Please, stick to the latter.

5. In Sections 3.7; Figure 8 should be placed at the end of the text in Line 334.

6. In Sections 3.8; Figure 9 should be placed at the end of the text in Line 372. Figures 9 should come before Figure 10. Put the legend of Figure 10 at the bottom.

7. Results and Discussion: Include some citations or refernces under each Section or compare with other works. There are missing citations in Sections 3.1; 3.2; 3.5 and 3.6; 

8. References: It seems the references format/style are not in accordance with the given 'Authors Instructions'. Please, refer to it, and revise accordingly. 

9. 'Sections' writing style/format: Refer to the given 'Authors Instructions' or published papers of the Journal.

Author Response

Reviewer 2:

The manuscript 'Simulation of Fluid Flow During Egg Pickling Under Different Inlet and Outlet Conditions in a Pulsed Pressure Tank with Liquid Circulation' meets scientific merit. The English is satisfactory and readers will understand its content. The authors provided information on four layers of egg pickling using Computational Fluid Dynamics by describing the algorithms and simulations involved. The research design and methods have been described appropriately. However, to maintain the quality impact of the Journal, the following comments are necessary for a revision:

1.Introduction: In Lines 83-85, provide the findings of those studies. What were the reported results? It is vital to mention them in the Introduction.

Response: Thanks very much for your feedback. The results of these studies on lines 83-85 are supplemented.as follows: Benni et al. compared the roof vent opening configurations by CFD simulations to optimize ventilation in the greenhouse. Cheng et al. studied the natural ventilation rates and airflow patterns in multi-span greenhouses and glass greenhouses, respectively. A good balance between airflow and wind speed.

  1. In Section 2.3; Figure 3(2) is not mentioned in the text. Please, highlight it in the text.

Response: Thanks for your useful suggestion. We are very sorry for some mistake that we made in this manuscript. We have checked and revised it. Figure 3(2) is already mentioned in Section 3.2.

  1. In Sections 3.4; Figure 6 should be placed at the end of the text in Line 279.

Response: Thanks. We are very sorry for some mistake that we made in this manuscript. We have checked and revised it.

4.In Sections 3.5; Figure 7 should be placed at the end of the text in Line 308. Ensure consistency of text format/style for the mention of Figs. 4 and 5 or Figure 4 and Figure. Please, stick to the latter.

Response: High appreciate the reviewer’s suggestions. We are very sorry for some mistake that we made in this manuscript. We have checked and revised it. To ensure the consistency of text format/style, Figs. 4 and 5 have been modified to Figure 4 and Figure 5.

5.In Sections 3.7; Figure 8 should be placed at the end of the text in Line 334.

Response: Thanks. We are very sorry for some mistake that we made in this manuscript. We have checked and revised it.

6.In Sections 3.8; Figure 9 should be placed at the end of the text in Line 372. Figures 9 should come before Figure 10. Put the legend of Figure 10 at the bottom.

Response: Thank you very much. We are very sorry for some mistake that we made in this manuscript. We have checked and revised it.

  1. Results and Discussion: Include some citations or references under each Section or compare with other works. There are missing citations in Sections 3.1; 3.2; 3.5 and 3.6.

Response: Thanks. We have added citations in Sections 3.1; 3.2; 3.5 and 3.6.

  1. References: It seems the references format/style are not in accordance with the given 'Authors Instructions'. Please, refer to it, and revise accordingly.

Response: Thanks very much. We are very sorry for some mistake that we made in this manuscript. We have checked and revised it.

  1. 9. 'Sections' writing style/format: Refer to the given 'Authors Instructions' or published papers of the Journal.

Response: Thanks. We are very sorry for some mistake that we made in this manuscript. We have checked and revised it.

Round 2

Reviewer 1 Report

Re-review of the manuscript entitled: Review of the manuscript entitled: Simulation of Fluid Flow During Egg Pickling Under Different Inlet and Outlet Conditions in a Pulsed Pressure Tank with Liquid Circulation

The reviewed manuscript is quite interesting. Simulations of the model of the improvement conditions of fluid flow in the pickling tank with a liquid-cycle system are an interesting and relatively difficult approach for correct modeling. 

Like I found the manuscript is corrected. The authors prepared adequate answers,  provide corrections, and improve the manuscript as well. The re-reviewed paper has novelty aspects.  The authors provided wider literature research. This will be of benefit to the recognition of the manuscript among other authors which concerned with similar cases. In summary: in view of the responses and prepared corrections, I recommend accepting the paper in the present form.